# A Life-Threatening Infection after Endobronchial Ultrasound Transbronchial Lung Biopsy with Guide Sheath: A Case Report

**DOI:** 10.3390/medicina58091275

**Published:** 2022-09-14

**Authors:** Insu Kim, Yeseul Oh, Min Ki Lee, Jung Seop Eom

**Affiliations:** 1Department of Internal Medicine, Dong-A University, College of Medicine, Busan 49201, Korea; 2Department of Internal Medicine, Pusan National University Hospital, Busan 49241, Korea; 3Biomedical Research Institute, Pusan National University Hospital, Busan 49241, Korea

**Keywords:** bronchoscopy, complication, extracorporeal membrane oxygenation, respiratory infection, ultrasonography

## Abstract

Background and Objectives: Endobronchial ultrasound transbronchial lung biopsy with guide sheath (EBUS-GS-TBLB) has been regarded as a reasonable diagnostic method with an acceptable diagnostic yield. In addition, EBUS-GS-TBLB is considered safer and less invasive compared to percutaneous needle biopsy and thoracoscopic surgery. However, we encountered a case of life-threatening procedure-related fatal infection, which was successfully managed. Case presentation: A 61-year-old man with a 30 pack-year smoking history was referred to our clinic with a necrotic lung mass in the right middle lobe on a chest computed tomography scan. EBUS-GS-TBLB was performed for a pathological diagnosis without immediate complications. Eight days after the procedure, the patient visited the hospital with sudden hemoptysis and severe dyspnea with fever. A chest computed tomography revealed a ruptured lung abscess and pneumonia, developed after EBUS-GS-TBLB. Extracorporeal membrane oxygenation (ECMO) and mechanical ventilation were initiated to manage refractory hypoxia. While maintaining ECMO, video-assisted thoracoscopic surgery was performed at the patient’s bedside in the intensive care unit. After surgery, the patient’s vital signs gradually improved, and a chest computed tomography revealed a reduction in the extent of the lung abscess. Results: Although EBUS-GS-TBLB is minimally invasive and relatively safe when used for the diagnosis of peripheral lung lesions, pulmonary physicians should be aware of this rare but critical complication. Conclusions: We suggest that the careful prescription of prophylactic antibiotics before EBUS-GS-TBLB would be wise if the mass featured a necrotic, cavitary, or cystic lesion.

## 1. Introduction

Transbronchial lung biopsy using endobronchial ultrasound transbronchial lung biopsy with guide sheath (EBUS-GS-TBLB) has become widely used for the histological diagnosis of peripheral lung lesions; the diagnostic performance is acceptable [1,2]. Generally, EBUS-GS-TBLB is considered safe, with overall complication rates of 0.8% (pneumothorax) and 0.5% (pulmonary infection) [3]. Thus far, most complications have been self-limiting, and fatal infections after the procedure are extremely rare [3,4]. Herein, we report a life-threatening infectious complication that developed after EBUS-GS-TBLB, which required management via extracorporeal membrane oxygenation (ECMO), mechanical ventilation (MV), and thoracoscopy in the intensive care unit (ICU).

## 2. Case Presentation

A 61-year-old man with a 30 pack-year smoking history and a persistent cough was referred to our clinic with a pulmonary lesion that was suspected to be lung cancer. A chest computed tomography revealed a round mass-like consolidation (5.9 cm in diameter) in the right middle lobe but no mediastinal lymph node enlargement (Figure 1A,B).

After the patient had been hospitalized, we used a 4.0 mm-diameter flexible bronchoscope (BF 260F; Olympus, Tokyo, Japan), a radial EBUS probe (UM-S20–17S; Olympus), and a guide sheath kit (K-201; Olympus) to perform EBUS-GS-TBLB [5]. The pathological diagnosis was squamous cell carcinoma; subsequently, a brain metastasis was found on brain magnetic resonance imaging, and the patient was finally diagnosed with stage IV non-small cell lung cancer. Chest radiographs were obtained 4 h after the procedure and on the next day; no pneumothorax or pneumonic consolidation was apparent. No antibiotics were prescribed during the hospital stay, and he was discharged the day after EBUS-GS-TBLB without any pain or fever. Five days later, he developed chills without fever, and an empirical oral antibiotic (cefditoren pivoxil) was prescribed at the outpatient clinic. Eight days after the procedure, he was admitted to the emergency department because of sudden hemoptysis, severe dyspnea with fever (38 °C), tachycardia, and tachypnea. A chest computed tomography revealed hydropneumothorax in the right middle lobe and pneumonic consolidation in both lower lobes, suggestive of pneumonia and a ruptured lung abscess that developed after EBUS-GS-TBLB (Figure 2A). The initial laboratory findings were as follows: white blood cell count, 16,330/μL; procalcitonin, 1.2 ng/mL; c-reactive protein, 31.1 mg/dL; fibrinogen, 787.3 mg/dL; d-dimer, 1.7 μg/mL; and fibrin degradation products, 5.1 μg/mL. A chest tube was urgently inserted into the right pleural space, and intravenous broad-spectrum antibiotics were prescribed in the emergency department.

Although oxygen was maximally supplied through a reservoir mask, the patient’s hypoxia persisted and his vital signs became unstable; ECMO and MV were initiated. Five days later, despite management with ECMO, MV, increased doses of vasopressor, and continuous renal replacement therapy in the ICU, the patient’s condition worsened, and his entire right lung became hazy (Figure 2B). On day 11 of ECMO, video-assisted thoracoscopic surgery was performed at the patient’s bedside in the ICU; purulent pus was evacuated and thick visceral pleura removed. After surgery, the patient’s vital signs gradually improved, and a chest computed tomography revealed a reduction in the extent of the lung abscess (Figure 2C).

Three days after surgery, the patient did not require ECMO; he was then weaned from the mechanical ventilator and tracheostomy. The patient was discharged 76 days after admission. He was found to exhibit a mutation in the epidermal growth factor receptor and was thus prescribed gefitinib. When comparing the chest X-rays and chest computed tomography scans obtained on discharge and three months later, a significant improvement was evident (Figure 3A–D).

## 3. Discussion

EBUS-GS-TBLB has been regarded as a reasonable diagnostic method with an acceptable diagnostic yield. Compared to percutaneous needle biopsy and thoracoscopic surgery, EBUS-GS-TBLB is safer and less invasive; it is typically performed with the patient under conscious sedation, induced by intravenous midazolam and fentanyl. The overall complication rate is reportedly 1.3–4.5%, and the incidence of infectious complications is reportedly 0.5–4.5% [2]. The development of a lung abscess after EBUS-GS-TBLB has been considered extremely rare (0.2–0.3%) [2]. However, we encountered a life-threatening infectious complication after EBUS-GS-TBLB. To the best of our knowledge, this is the first report of a potentially fatal procedure-related complication after EBUS-GS-TBLB; here, we managed the complication by means of ECMO, continuous renal replacement therapy, MV, and thoracoscopic surgery in the ICU.

The British Thoracic Society guidelines for bronchoscopy do not recommend the use of prophylactic antibiotics to prevent endocarditis, fever, or pneumonia before biopsy using bronchoscopy [6]. Additionally, in the study by Souma et al., it is known that the use of antibiotics to prevent complications related to the procedure is not beneficial [7]. In addition, the study by Shimoda also suggests that the use of prophylactic antibiotics to prevent complications related to the bronchoscopic procedure cannot be determined [8]. Therefore, the use of antibiotics to prevent infection complications related to the procedure is not beneficial, and the procedure should be performed more carefully for certain conditions that are predicted to have a high incidence of infection after the procedure. The conditions that can cause high infection complications after a bronchoscopic biopsy are presented as cavitation in the lesion, intratumoral low-density areas in the lesion, and responsible bronchus stenosis in the study by Souma [7], and as necrosis and/or a cavity in the tumor, tumor diameter >30 mm, and serum albumin <4.0 g/dL in the study by Shimoda [8]. In our case report, the patient had a high probability of infection with a mass exceeding 30 mm in diameter with necrosis at the center of the tumor, reflecting the above study results. Souma and others suggest frequent follow-ups to prevent infection complications after the procedure [7], but there are practical limitations, and as the authors recognize, this is not a definitive strategy. On the other hand, Shimoda et al [8]. suggested that the use of antibiotics that can cover oral flora and anaerobic bacteria may be necessary for high-risk patients [8]. Therefore, the use of prophylactic antibiotics cannot be excluded from groups at a high risk of postoperative infectious complications. We hope that further prospective studies will identify prophylactic antibiotics that may prevent post-procedure complications in patients undergoing EBUS-GS-TBLB.

## 4. Conclusions

We describe a patient who developed a life-threatening lung infection as a complication of EBUS-GS-TBLB and who struggled to recover despite ECMO, continuous renal replacement therapy, MV, and thoracoscopic surgery in the ICU. Although EBUS-GS-TBLB is minimally invasive and relatively safe when used for the diagnosis of peripheral lung lesions, pulmonary physicians should be aware of this rare but critical complication.

## Figures and Tables

**Figure 1 medicina-58-01275-f001:**
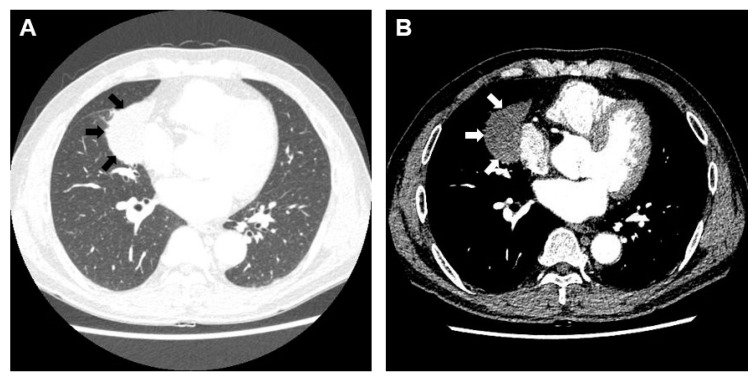
Initial Computed Tomography (CT) findings of a patient with lung cancer. (**A**) Chest CT revealed a mass-like consolidation (59 × 35 mm) in the right middle lobe (black arrow). (**B**) In the mediastinal window, a low-density area was visible in the target lesion, without cavity formation (white arrow).

**Figure 2 medicina-58-01275-f002:**
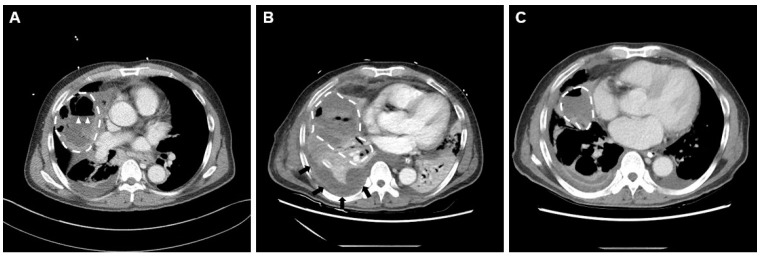
Chest CT findings after lung abscess development. (**A**) A CT scan taken in the emergency department revealed a lung abscess (broken lines) and an air-fluid line (arrowhead) in the right middle lobe. (**B**) A chest CT scan taken before surgery revealed abscess deterioration (broken lines), pleural effusion (black arrow), and internal necrosis. (**C**) A CT scan taken after surgery revealed improvement of the lung abscess (broken lines) and other features.

**Figure 3 medicina-58-01275-f003:**
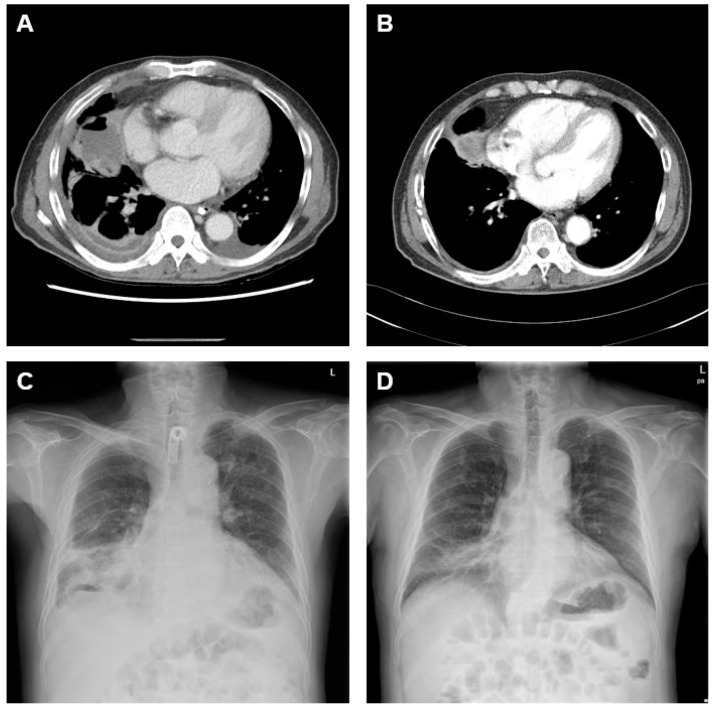
Serial changes in the patient’s chest CT and X-ray findings after discharge. A comparison of CT scans taken (**A**) at discharge and (**B**) 3 months later revealed significant improvement. A comparison of the chest X-rays taken (**C**) at discharge and (**D**) 3 months later showed that the abscess had disappeared.

## Data Availability

Not applicable.

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
