# Peer review of "A Life-Threatening Infection after Endobronchial Ultrasound Transbronchial Lung Biopsy with Guide Sheath: A Case Report"

_medicina, 2022, doi:10.3390/medicina58091275_

Round 1

Reviewer 1 Report

This case report describes a case of life-threatening lung abscess after EBUS-GS-TBLB, which was managed by ECMO, continuous renal replacement therapy, MV, and thoracoscopic surgery in the ICU. And this manuscript is concise and well written.

Lung abscess after EBUS-GS-TBB is a very rare but fatal complication that clinicians commonly experience. Thus, it lacks novelty.

As stated in the text, as a clinician, I agree with prophylactic antibiotics treatment for patients at high risk for lung abscess, but the evidence is lacking at this time.

There have already been several reports that have discussed this concern. Some recent reports are listed below. 

Souma T, et al. Chest. 2020 Aug;158(2):797-807.

Shimoda M, et al. J Infect Chemother. 2021 Feb;27(2):237-242.

These should be referred to and the risk of this case reevaluated in detail.

Lines 115-133

In risk assessment and prophylactic antibiotics treatment, the discussion in EBUS-TBNA and EUS-FNA is not applicable.

It is not at all clear that this conclusion can be drawn from this case report.

Minor comments

Line 56

BF-260 has a 4.9 mm diameter; BF-P260F has a 4.0 mm diameter.

Figure legend Figure 1

Please spell out CT.

Author Response

â–ª Major Comment

 This case report describes a case of life-threatening lung abscess after EBUS-GS-TBLB, which was managed by ECMO, continuous renal replacement therapy, MV, and thoracoscopic surgery in the ICU. And this manuscript is concise and well written.

Lung abscess after EBUS-GS-TBB is a very rare but fatal complication that clinicians commonly experience. Thus, it lacks novelty.

 As stated in the text, as a clinician, I agree with prophylactic antibiotics treatment for patients at high risk for lung abscess, but the evidence is lacking at this time.

There have already been several reports that have discussed this concern. Some recent reports are listed below.

Souma T, et al. Chest. 2020 Aug;158(2):797-807.

Shimoda M, et al. J Infect Chemother. 2021 Feb;27(2):237-242.

These should be referred to and the risk of this case reevaluated in detail.

Lines 115-133

In risk assessment and prophylactic antibiotics treatment, the discussion in EBUS-TBNA and EUS-FNA is not applicable.

It is not at all clear that this conclusion can be drawn from this case report.

â–ª Response

We thank you for presenting good clinical materials. We reviewed the presented data and determined that the content of prophylactic antibiotics using needle aspiration methods such as EBUS-TBNA was not suitable for this case. Therefore, Lines 115-133 were removed and the contents of the presented bronchoscopic biopsy or EBUS-TBLB related infection were added in the Discussion part as follows (see page 4, line 113): “The British Thoracic Society guidelines for bronchoscopy do not recommend the use of prophylactic antibiotics to prevent endocarditis, fever, or pneumonia before biopsy us-ing bronchoscopy [6]. Also, in the study of Souma et al., it is known that the use of antibiotics to prevent complications related to the procedure is not beneficial [7]. In addition, the study by Shimoda also suggests that the use of prophylactic antibiotics to prevent complications related to the bronchoscopic procedure cannot be determined [8]. Therefore, the use of antibiotics to prevent infection complications related to the procedure is not beneficial, and the procedure should be performed more carefully for certain conditions that are predicted to have a high incidence of infection after the procedure. The conditions that can cause high infection complications after the bronchoscopic biopsy are presented as cavitation in the lesion, intratumoral low-density areas in the lesion, responsible bronchus stenosis in the study of Souma [7], and necrosis and/or a cavity in the tumor, tumor diameter >30mm, and serum albumin <4.0 g/dL in the study of Shimoda [8]. Reflecting the above study results, in our case report, the patient had a high probability of infection with a mass exceeding 30 mm in diameter with necrosis at the center of the tumor. Souma and others suggest frequent follow-up to prevent infection complications after the procedure [7], but there are practical limitations, and as the authors recognize, it is not a definitive strategy. On the other hand, Shimoda et al. suggested that the use of antibiotics that can cover the oral flora and anaerobic bacteria may be necessary for high-risk patients [8]. Therefore, the use of prophylactic antibiotics cannot be excluded from the high-risk group of postoperative infectious complications. We hope that further prospective study will identify prophylactic antibiotics that may prevent post-procedure complications in patients undergoing EBUS-GS-TBLB.”

â–ª Minor Comment 1

Line 56

BF-260 has a 4.9 mm diameter; BF-P260F has a 4.0 mm diameter.

â–ª Response

We thank the reviewer for bringing this point to our attention. As suggested by the editor, we have changed the specification of bronchoscopy in the Case presentation as follows (see page 2, line 56): “BF 260F”

â–ª Minor Comment 2

Figure legend Figure 1

Please spell out CT.

â–ª Response

We thank the reviewer for bringing this point to our attention. As suggested by the editor, we have added a full spelling of CT in Figure 1 of the Case presentation as follows (see page 2, line 52): “Initial Computed Tomography (CT) findings of a patient with lung cancer.”

Reviewer 2 Report

Dear Authors,

This case is very interesting with clinical significance. Authors should elaborate more about why prophylactic antibiotics prescription is necessary. 

Author Response

# Response to the comments of Reviewer 2

â–ª Comment

 This case is very interesting with clinical significance. Authors should elaborate more on why prophylactic antibiotics prescription is necessary.

â–ª Response

We thank the reviewer for bringing this point to our attention. In this case, a bronchoscopy biopsy was performed on a patient who has a mass with internal necrosis greater than 30 mm in diameter. Therefore, prophylactic antibiotics were used as it was evaluated as a high-risk group for the occurrence of procedure-related infectious complications. Therefore, the criteria for the high-risk group and the use of prophylactic antibiotics for the postoperative complications were added to the Discussion part as follows (see page 4, line 123-135): “The conditions that can cause high infection complications after the bronchoscopic biopsy are presented as cavitation in the lesion, intratumoral low-density areas in the lesion, responsible bronchus stenosis in the study of Souma [7], and necrosis and/or a cavity in the tumor, tumor diameter >30mm, and serum albumin <4.0 g/dL in the study of Shimoda [8]. Reflecting the above study results, in our case report, the patient had a high probability of infection with a mass exceeding 30 mm in diameter with necrosis at the center of the tumor. Souma and others suggest frequent follow-up to prevent infection complications after the procedure [7], but there are practical limitations, and as the authors recognize, it is not a definitive strategy. On the other hand, Shimoda et al. suggested that the use of antibiotics that can cover the oral flora and anaerobic bacteria may be necessary for high-risk patients [8]. Therefore, the use of prophylactic antibiotics cannot be excluded from the high-risk group of postoperative infectious complications.”

References

  1. 2020 Aug;158(2):797-807.
  2. J Infect Chemother. 2021 Feb;27(2):237-242.

Reviewer 3 Report

It is difficult to establish if this case report s related to the EBUS-GS-TBLB or it is a complication in the evolution of the tumor with abcess and fistulisation on the pleura. I suggest to add also this possibility of the mechanism of cthe omplication.

Author Response

# Response to the comments of Reviewer 3

â–ª Comment

 It is difficult to establish if this case report s related to the EBUS-GS-TBLB or it is a complication in the evolution of the tumor with abscess and fistulisation on the pleura. I suggest to add also this possibility of the mechanism of the complication

â–ª Response

We thank the reviewer for bringing this point to our attention. Factors related to the occurrence of complications include host factors, factors related to procedures, and factors related to pathogens. In a study by Shimoda et al., it is known that a large number of biopsies for the recently required molecular and immunological tests could weaken the mucosal barrier and affect infection. In the study of Ishida et al., it is known that the occurrence of complications is determined by the factors the patient has (size of the mass, whether it is internal necrosis or cavity). Therefore, the mechanism of infection has not been established at present, and additional prospective or experimental studies are needed.

References

  1. Respir Investig. 2015 May;53(3):129-32.
  2. 2020 Aug;158(2):797-807.
  3. J Infect Chemother. 2021 Feb;27(2):237-242.
  4. 2020 Aug;158(2):458-460.

Round 2

Reviewer 1 Report

I am satisfied that my queries and concerns have been addressed by the authors' revisions. The manuscript is well disccussed and more balanced.